# Modeling Therapy-Driven Evolution of Glioblastoma with Patient-Derived Xenografts

**DOI:** 10.3390/cancers14225494

**Published:** 2022-11-09

**Authors:** Matthew McCord, Elizabeth Bartom, Kirsten Burdett, Aneta Baran, Frank D. Eckerdt, Irina V. Balyasnikova, Kathleen McCortney, Thomas Sears, Shi-Yuan Cheng, Jann N. Sarkaria, Roger Stupp, Amy B. Heimberger, Atique Ahmed, Charles David James, Craig Horbinski

**Affiliations:** 1Department of Pathology, Northwestern University Feinberg School of Medicine, Chicago, IL 60611, USA; 2Department of Biochemistry and Molecular Genetics, Northwestern University Feinberg School of Medicine, Chicago, IL 60611, USA; 3Department of Preventive Medicine, Northwestern University Feinberg School of Medicine, Chicago, IL 60611, USA; 4Robert H. Lurie Comprehensive Cancer Center, Northwestern University Feinberg School of Medicine, Chicago, IL 60611, USA; 5Lou and Jean Malnati Brain Tumor Institute of Northwestern Medicine, Chicago, IL 60611, USA; 6Department of Neurological Surgery, Northwestern University Feinberg School of Medicine, Chicago, IL 60611, USA; 7Department of Neurology, Northwestern University Feinberg School of Medicine, Chicago, IL 60611, USA; 8Department of Radiation Oncology, Mayo Clinic, Rochester Minnesota, Rochester, MN 55905, USA

**Keywords:** CNS cancers, gliomas, glioblastomas, tumor evolution, DNA damage and repair, chemotherapy, drug resistance, preclinical models, xenograft models

## Abstract

**Simple Summary:**

Glioblastoma (GBM) is the most common and aggressive adult-type diffusely infiltrating glioma. These tumors invariably develop resistance to standard treatment with radiation and temozolomide, leading to recurrence and almost always fatal outcomes. In vivo models of such recurrences are limited, and new therapies for recurrent GBM are usually tested on therapy-naïve preclinical models, which do not accurately predict outcomes in clinical trials. Experimental therapies which are effective against therapy-naïve tumor models in mice often fail to achieve survival benefit in patients with recurrent, therapy-resistant GBMs. In this study, we developed multiple treatment-resistant GBM models by exposing patient-derived xenografts (PDX) of GBM to radiation and temozolomide. These therapy-resistant PDX reflect key genetic and phenotypic features of recurrent GBM in patients. These PDX models are stable and expandable, and can serve as a valuable tool for testing new therapies in a setting that more accurately models GBM that recurs after front-line therapy.

**Abstract:**

Adult-type diffusely infiltrating gliomas, of which glioblastoma is the most common and aggressive, almost always recur after treatment and are fatal. Improved understanding of therapy-driven tumor evolution and acquired therapy resistance in gliomas is essential for improving patient outcomes, yet the majority of the models currently used in preclinical research are of therapy-naïve tumors. Here, we describe the development of therapy-resistant IDH-wildtype glioblastoma patient-derived xenografts (PDX) through orthotopic engraftment of therapy naïve PDX in athymic nude mice, and repeated in vivo exposure to the therapeutic modalities most often used in treating glioblastoma patients: radiotherapy and temozolomide chemotherapy. Post-temozolomide PDX became enriched for C>T transition mutations, acquired inactivating mutations in DNA mismatch repair genes (especially *MSH6*), and developed hypermutation. Such post-temozolomide PDX were resistant to additional temozolomide (median survival decrease from 80 days in parental PDX to 42 days in a temozolomide-resistant derivative). However, temozolomide-resistant PDX were sensitive to lomustine (also known as CCNU), a nitrosourea which induces tumor cell apoptosis by a different mechanism than temozolomide. These PDX models mimic changes observed in recurrent GBM in patients, including critical features of therapy-driven tumor evolution. These models can therefore serve as valuable tools for improving our understanding and treatment of recurrent glioma.

## 1. Introduction

Adult-type diffuse gliomas affect more than 20,000 patients annually in the US and are nearly always incurable [1,2]. IDH wildtype glioblastoma (GBM), the most common and aggressive form of adult-type diffuse glioma, has a median survival of only 15 months, with less than 5% of patients surviving five years, even when treated with aggressive multi-modal therapy [3,4]. The DNA alkylating chemotherapeutic agent temozolomide (TMZ) provides some limited survival benefit [3,5], but tumors invariably develop resistance to TMZ and recur [6,7]. TMZ-induced cytotoxicity depends on tumor cell DNA mismatch repair (MMR) enzymes: MSH2, MSH6, MLH1, and PMS2. When the enzyme complexes cannot repair base mismatches between thiamine and TMZ-alkylated guanine, the MMR enzymes help initiate cellular apoptosis [8,9].

Because MMR enzymes not only repair TMZ-induced DNA damage but also trigger apoptosis when that damage cannot be repaired, a common mechanism of TMZ resistance is inactivating mutations in those DNA MMR genes, especially *MSH6*, resulting in defective DNA mismatch repair, increased cellular tolerance of base mismatches, and lower sensitivity to TMZ [10,11,12,13,14,15]. Post-TMZ treatment, recurrent gliomas develop a distinct mutation profile, designated “Signature 11,” which is enriched for cytosine to thymine (C>T) transitions due to accumulation of thymine residues mismatched with TMZ-alkylated guanine [16]. When TMZ-induced MMR deficiency develops, tumors often develop elevated tumor mutation burden (TMB). Such hypermutated gliomas are often defined as having ≥10 mutations per megabase (Mb) of DNA, as compared to non-hypermutated gliomas that typically have ~1 mutation/Mb [10,13,17,18,19]. Clinically this suggests that subsequent treatment with TMZ, or other DNA-damaging alkylating agents, would be unsuccessful. However, hypermutated tumors might be more amenable to immunotherapy [20,21,22,23,24,25], although this is controversial in gliomas [26,27,28].

Because TMZ is used in the post-surgical treatment of most GBM patients, models of acquired TMZ resistance can improve our understanding of how to best treat recurrent GBM. However, most of the in vitro and in vivo models used in preclinical research are treatment-naïve GBM. Accurate models of post-therapy, recurrent tumors are therefore needed. Here, we describe the development and evaluation of post-therapy GBM PDX models that have been characterized for TMB and whole exome mutation profiles, including MMR mutations.

## 2. Materials and Methods

### 2.1. Cell Lines and Cell Culture

Patient-derived GBM xenografts were established from treatment-naïve, IDH-wildtype GBM: GBM6 (*MGMT* promoter unmethylated), GBM12 (*MGMT* promoter methylated), and GBM43 (*MGMT* promoter unmethylated). These cell lines were obtained from the laboratory of Dr. Jann N. Sarkaria. Xenografts propagated as subcutaneous tumors were used as cell sources for establishing intracranial tumors, as previously described [29,30,31].

### 2.2. Intracranial Engraftment of PDX Cells

The overall process for developing treated derivative PDX from treatment-naïve PDX is summarized in Figure 1A. PDX, whose cells had been stably modified with a luciferase reporter for use with bioluminescence imaging (BLI) [30,31] were grown subcutaneously in adult athymic nude mice, and resected after animals were euthanized. Parental PDX were originally developed in athymic nude mice, and the same host species was used in developing and testing the derivative PDX. Cell suspensions were prepared from resected tumors as previously described [29]. A total of 3 × 10^5^ cells from cell suspensions were injected intracranially into athymic nude mice, using injection coordinates previously indicated [31]. Successful tumor engraftment and progressive growth was confirmed by serial BLI (IVIS Spectrum, Perkin Elmer), as illustrated in Figure 1B,C, and as described previously [30].

### 2.3. In Vivo Treatment of Parental PDX

Seven days following intracranial injection of tumor cells and confirmation of progressive tumor growth, cohorts of mice received one of the following treatment regimens: radiotherapy (RT) only (2 Gy/day for 5 consecutive days), TMZ only (10 mg/kg/day via oral gavage for 5 consecutive days), or concurrent RT and TMZ therapy using the indicated monotherapy regimens. Some of the TMZ-treated mice received additional TMZ treatment on indication of tumor re-growth from initial therapy as indicated by twice weekly BLI. Treatment regimens are summarized in Table 1. Additional details, including time points of all TMZ cycles, are available in Appendix A.

### 2.4. Propagation of Derivative PDX

Mice with intracranial tumors were euthanized upon becoming symptomatic. Brains were immediately resected and visible tumor dissected. Dissected tumors were converted to a cell suspension, with 200 μL of suspended cells (10^5^ cells/μL) injected subcutaneously into a new host mouse. After growth to a volume of 2 cm^3^, the mouse bearing subcutaneous tumor was euthanized, with tumor immediately resected and again converted to a cell suspension which was subcutaneously injected into a new host mouse. The cycle of subcutaneous propagation was repeated once more (three subcutaneous passages in total). Third passage subcutaneous tumors were harvested, with portions of each snap frozen for molecular analysis and cryopreserved for subsequent use for in vitro and in vivo experiments. Unique parental PDX (3 total) and treatments for derivative PDX (21 total) are summarized in Table 1.

### 2.5. Cell Viability Assays

Cell viability was assessed using the MTT assay. Cells were plated at 3000 cells per well in a 96-well plate with four replicates plated for each condition. After 24 h, media was changed to media supplemented with TMZ (50, 100, 250, 500, and 1000 μM). Media supplemented with DMSO was used as vehicle control. After 72 h of incubation, cells were washed with phosphate-buffered saline (PBS) and incubated with MTT reagent (Thermo Fisher Scientific, Waltham, MA, USA) diluted 10% in regular culture media. Cells were then maintained for 4 h at 37 °C, after which MTT reagent was removed and samples resuspended in DMSO, with sample absorbance of each determined using a plate reader (BioTek, Winooski, VT, USA, Synergy 2), and results tabulated per standard protocols [32].

### 2.6. Testing Parental and Derivative PDX for Response to Therapy

To establish subcutaneous tumors, 5 × 10^6^ cryopreserved cells from either parental or derivative PDX were injected into the flanks of adult athymic nude mice. Mice bearing subcutaneous tumors were euthanized, with tumors resected and used to prepare cell suspensions for intracranial injection as previously described [29]. Intracranial tumor engraftment and growth were monitored via BLI. One week following intracranial injection of tumor cells, cohorts of mice received either a single cycle of TMZ or RT. In one experiment, a cohort of mice with recurrent post-treatment tumor received a single dose of 1-(2-Chloroethyl)3-cyclohexyl-1-nitrosourea (CCNU, also known as lomustine) (50 mg/kg via oral gavage). Mice were assigned randomly to treatment or vehicle control groups.

### 2.7. Whole Exome Sequencing

Prior to library preparation, genomic DNA (gDNA) was quantified by Qubit and assessed for quality on an Agilent Bioanalyzer. For library construction, Illumina TruSeq Exome Library Prep Kit was employed for all steps of the library prep process. The gDNA was fragmented to 150 base pair (bp) insert size using Covaris shearing, followed by end repair, library size selection, and 3′ end adenylation. Multiple indexing adapters were then ligated to the ends of the DNA fragments. A limited-cycle-number PCR was used to selectively enrich for DNA fragments with ligated adapters for library development. After being validated with Qubit and Agilent Bioanalyzer, DNA libraries carrying unique barcoding indexes were pooled and hybridized to exome oligo probes to capture the exonic regions of the genome. This capture process was conducted twice to ensure high exome specificity. Captured libraries were amplified with an 8-cycle PCR. After post-PCR purification, enriched libraries were validated with Qubit quantification and Bioanalyzer quality check using a High Sensitivity DNA chip. The sequencing of the libraries was conducted on Illumina NextSeq 500 (PE75) and HiSeq 4000 (PE100) sequencers with dual indexing.

### 2.8. Bioinformatics and Biostatistics Analyses

FastQ files were first assessed for quality with FastQC, available at https://bioinformatics.babraham.ac.uk/projects/fastqc (accessed on 31 August 2015), and validated with FastQ_screen and NGS checkmate to ensure accuracy of sample metadata. Paired-end reads were then trimmed with Trimmomatic v0.33 to remove low-quality or adapter sequence and aligned to human genome reference assembly hg38 with bwa mem. PCR duplicates were marked with SAMBLASTER. GATKv3.6 was used to realign reads and recalibrate quality scores in the aligned BAM files. Variant calls were made using GATK’s Mutect2 to identify variants in each treated PDX relative to its untreated parental PDX.

Differences between mean values of groups were compared using unpaired t-test, one way ANOVA (two groups) or two-way ANOVA with Tukey’s multiple comparisons (multiple groups). Kaplan–Meier curves were used to illustrate differences in overall survival (OS), while comparison between groups was performed using log-rank tests. Differences between observed and expected outcomes were compared with Fisher’s exact test. Simple linear regression was performed comparing coding sequence (CDS) length to number of mutations. For all statistical tests, *p* values less than 0.05 were considered significant. Figures were generated, and statistical analyses were performed with GraphPad PRISM 5 software and the R statistical environment version 4.0.2 along with extension packages VariantAnnotation (v 1.36.0), VennDiagram (v 1.6.20), and MutationalPatterns (v 3.0.1) [33,34,35].

## 3. Results

### 3.1. Establishment, Growth, and Primary Treatment of Parental PDX

Representative bioluminescence (BLI) studies of engraftment and treatment of GBM6 PDX in 5 mice are shown in Figure 1B,C. A mouse engrafted with parental GBM6 and treated with vehicle control showed rapid tumor growth beginning at day 40. Treatment of engrafted GBM6 with RT monotherapy (days 27–31 post-engraftment) delayed tumor growth until days 55–60. Monotherapy with a cycle of TMZ (days 27–31 post-engraftment) delayed tumor growth until days 85–90 (Figure 1B). Combined therapies delayed tumor growth even more effectively, as RT with a concomitant cycle of TMZ (days 27–31 post-engraftment) delayed tumor growth until day 110 (Figure 1C). Tracking of BLI on a mouse receiving therapy with RT and multiple cycles of TMZ showed gradual development of resistance with each successive TMZ cycle. The mouse received concomitant RT and TMZ (days 27–31 post-engraftment) which delayed tumor growth until day 90. A second cycle of TMZ was given (days 99–103) and BLI showed tumor regression. BLI showed tumor re-growth beginning at day 148, and a third cycle of TMZ was given (days 155–159). Regression did not occur after the third cycle. Rather, tumor growth accelerated from days 159–167. This tumor was harvested and became PDX derivative m3378.

This experimental approach was used in developing additional treated derivative PDX from parental GBM6, GBM12, and GBM43 PDX, with specific treatment regimens for each derivative PDX summarized in Table 1 (further details in Appendix A).

### 3.2. TMZ Treatment Results in Marked Increase in Tumor Mutation Burden (TMB) and a Distinct Mutation Profile

After three consecutive passages as subcutaneous tumors, to ensure the stability of acquired mutations, whole exome sequencing (WES) was done on PDX from each treatment group, including parental untreated PDX (Table 1, Appendix A). A single cycle of TMZ completely eliminated GBM43 PDX, so most subsequent analyses focused on GBM6 and GBM12. There was no overlap in genes mutated among the three GBM6 PDX treated with RT alone (Figure 2A), and only 1–3 common genes were mutated among all GBM6 PDX after at least one cycle of TMZ (Figure 2B,C). As has been demonstrated in GBM patients [10,16], C>T transitions in GBM6 PDX were greatly enriched after treatment with TMZ alone or RT+TMZ, but not RT alone (Figure 2D and Appendix A).

GBM12 had 53 common genes mutated after RT alone (Figure 3A), and 13–20 common genes mutated after various exposures to either TMZ monotherapy or RT+TMZ (Figure 3B,C). Like GBM6, GBM12 PDX showed much higher rates of C>T transitions associated with TMZ exposure, but not RT alone (Figure 3D and Appendix A). Like GBM6 and GBM12, GBM43 PDX treated with RT alone did not show substantial enrichment of C>T mutations (Appendix A).

Among all PDX, TMB increased with increasing cycles of TMZ. GBM6 gained up to 65.98 mutations per megabase (Mb) after 3 cycles of TMZ, and GBM12 gained up to 151.85 after 4 TMZ cycles. Along with that increase in TMB, DNA MMR mutations gradually emerged, most often involving *MSH6* but also including *MSH2*, *MLH1*, and *PMS2* after repeated exposures to TMZ (Table 1). Derivative PDX with at least one DNA MMR gene mutation showed average TMB gain of 55.45/Mb, compared to 9.54 for derivatives with intact DNA MMR genes (*p* = 0.0053, Figure 4A). Derivative PDX with mutations in 2 or more DNA MMR genes showed average TMB gain of 107.9/Mb, compared to 16.13 for derivatives with mutation in only one gene, and 9.54 for MMR-intact derivatives (*p* < 0.0001 for both comparisons, Figure 4B). Overall, 7 out of the 14 derivative PDX treated with TMZ developed MMR mutations, while none of the 7 treated only with RT developed MMR mutations (*p* = 0.047).

The most frequently mutated gene in TMZ-treated PDX was *TTN*, encoding a muscle protein called titin (Figure 4C and Appendix A). *TTN* also has the longest coding sequence of any human gene: 283,000 bp [36], meaning that any mutation-inducing agent would statistically be expected to alter longer genes like *TTN* more frequently. Indeed, longer coding sequence length strongly correlated with the likelihood of mutation (slope = 8.51 × 10^−6^, R^2^ = 0.4666, *p* < 0.0001, Figure 4D), even when the *TTN* outlier was eliminated (slope =1.71 × 10^−4^, R^2^ = 0.3030, *p* = 0.0007, Figure 4E). In contrast, *MSH6* is only the 1458th longest gene at 4083 bp, yet was the eighth-most frequently mutated gene in TMZ-treated PDX (*MACF1* was another outlier, 22,779 bp, encoding microtubule actin crosslinking factor 1). As expected, PDX treated only with RT showed no preferential mutation of genes with long CDS (Figure 4F,G).

### 3.3. Patterns of Therapy Response in Post-Treatment GBM PDX

Derivative PDX were tested for response to the same treatment regimens used in their initial development. Within GBM6 PDX, mice engrafted with the post-RT+TMZ m3378 derivative (*MSH6* and *PMS2* mutant, TMB gain of 65.98) experienced the shortest median overall survival in response to additional TMZ (42 days), followed by m4051 at 49 days (*MSH6* mutant, TMB gain of 20.62/Mb), m4082 at 74 days (MMR intact, TMB gain of 5.98), and parental GBM6 at 80 days (MMR intact) (Figure 5A). A similar pattern held in vitro, wherein the *MSH6*-mutant m3378 and m4051 derivatives showed lower sensitivity to TMZ than parental GBM6 (Figure 5B,C). Maximal effect for parental GBM6 resulted in a 73.7% reduction in relative cell viability at 1000 µM, whereas maximal effect for m3378 resulted in only 51.1% reduction, and m4051 only in 38.1% reduction at the same concentration. Post-TMZ GBM12 m2671 PDX (*MSH6* mutant, TMB gain of 22.36) also showed a weaker response to in vivo TMZ than its treatment-naïve, MMR-intact parental counterpart, as indicated by shorter median survival of engrafted mice treated with additional TMZ (54 versus 61 days, *p* = 0.002, Figure 5D).

While post-TMZ, MMR-mutant GBM6 m3378 and m4051 PDX were resistant to additional TMZ, both became slightly more sensitive to RT than parental GBM6 (median survival 50–51 days versus 42.5 days, *p* < 0.01, Figure 5E). The m4056 derivative, which had previously been exposed to only one round of TMZ and was MMR-intact, was the most responsive to RT (median survival 64 days, *p* < 0.01 versus all other groups).

CCNU is a DNA-crosslinking nitrosourea that has been used in treating GBM, especially in the setting of recurrent tumor [37,38]. To determine whether acquired TMZ resistance also confers resistance to CCNU, mice engrafted with GBM6 PDX derivative m3378 were treated with CCNU. All m3378-engrafted mice treated with CCNU experienced complete tumor regression, with 100% survival; in contrast, all vehicle- or TMZ-treated m3378 mice died within 52 days after engraftment (*p* < 0.01, Figure 5F). Postmortem microscopic analysis of CCNU-treated mice showed no evidence of residual GBM (not shown).

## 4. Discussion

Primary brain cancers like IDH wild-type GBM account for more years of life lost, on average, than any other form of cancer, highlighting the need for new and improved therapies [39]. GBM recurrence following TMZ therapy represents a major barrier to survival improvement, and is a topic of ongoing investigation [40]. Representative models of recurrent GBM are needed for improved understanding and treatment of these tumors. Experimental therapies for GBM that show promise in therapy-naïve PDX models in preclinical studies (e.g., Cediranib, Bevacizumab), have usually failed to improve overall survival for patients with recurrent GBM in clinical trials [41,42,43,44].

Through repeated in vivo TMZ exposure, we developed GBM PDX models which display key genotypes and phenotypes associated with recurrent GBM in patients. These models mimic many of the key aspects of post-TMZ GBM in patients, including C>T transition enrichment, DNA MMR mutations (especially in *MSH6*), increased TMB, and TMZ resistance, that have been reported in other experimental and patient-based studies [10,11,12,13,14,16,45]. Consequently, they should prove valuable for research aimed at treating TMZ-resistant GBM. Previous work has successfully established PDX from post-therapy, recurrent GBM, on a more limited basis than primary GBM PDX [46]. Our approach involving serial in vivo treatment of therapy-naïve PDX has the advantage of giving investigators complete control over treatment modality and intensity.

Early data from these post-TMZ GBM PDX models not only show that they mimic patient GBM responses to front-line TMZ, but that they can also accurately model responses to second-line therapies when GBMs recur. First, we found that treatment of GBM PDX with TMZ can enhance subsequent RT response, even when several passages occur between when a PDX is treated with TMZ and RT. Others had previously shown that TMZ treatment increases glioma cell radiosensitivity [47,48]. Second, TMZ-treated PDX that developed TMZ-resistance, MMR mutations, and hypermutation remained highly sensitive to CCNU in vivo. Unlike TMZ, CCNU can induce apoptosis in tumor cells with defective DNA MMR [49], and has been suggested as therapy for patients with hypermutated, MMR-deficient gliomas [50]. Our findings match prior data by others, in which GBM cells derived from post-TMZ, MMR-mutant patient tumors were TMZ resistant but sensitive to CCNU, and TMZ-naïve GBM cells with artificial MMR gene inactivation acquired TMZ resistance while remaining sensitive to CCNU [10]. Our findings also match the observation that hypermutation rarely develops in gliomas post-CCNU therapy [10,51], and that CCNU might even be useful in conjunction with TMZ as front-line therapy in GBM [37].

Our work suggests a PDX application that has been underutilized—the longitudinal study of tumor evolution during therapy. This concept has gained traction with the Glioma Longitudinal AnalySiS (GLASS) consortium [52,53,54], and highlights another advantage of our approach to developing therapy-resistant PDX. By engrafting several mice with a single tumor cell source, one can harvest the same tumor at different stages of treatment in order to identify temporal aspects of therapy-driven tumor evolution, including changes in tumor microenvironment [55]. The results from such studies could prove informative with regard to the length of time a specific treatment remains effective, as well as potential therapeutics to use once the initial therapy has become ineffective. Because our post-treatment PDX derivatives retained their molecular and phenotypic characteristics even after several passages without any additional treatment, they appear to be stable, and are therefore amenable to expansion and distribution among numerous laboratories.

## 5. Conclusions

The post-therapy PDX described here show key features known to arise in treated patient GBMs, and are therefore useful models for developing better ways of managing recurrent GBM. This approach could also serve as a paradigm for developing and studying mechanisms of therapy resistance in other tumor types.

## Figures and Tables

**Figure 1 cancers-14-05494-f001:**
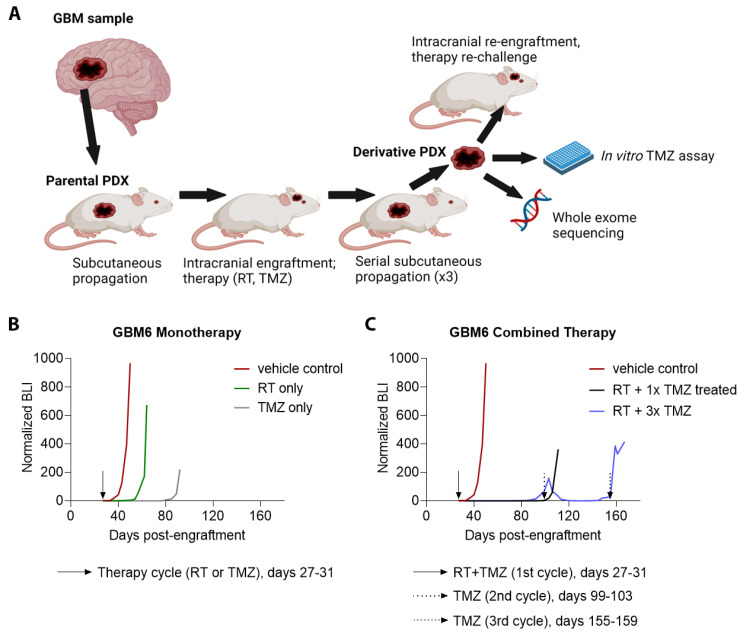
Establishment and primary treatment of parental PDX. (**A**) Illustration of workflow. Patient-derived tumor cell lines were propagated subcutaneously in athymic nude mice, then engrafted intracranially and treated with radiation (RT) and temozolomide (TMZ). Post-therapy PDX were serially passaged subcutaneously through 3 mice. After therapy and propagation, derivative PDX cells were tested for in vitro and in vivo therapy resistance and underwent whole-exome sequencing (WES). (**B**) Bioluminescence (BLI) monitoring of intracranial engraftment of parental GBM6 with vehicle treatment versus treatment with RT monotherapy, and versus TMZ monotherapy. (**C**) BLI monitoring of intracranial engraftment of parental GBM6 with vehicle treatment versus treatment with one concomitant cycle of RT and TMZ, and versus treatment with a concomitant cycle of RT and TMZ followed by two additional cycles of TMZ at the indicated time points (*n* = 5 mice total for (**B**,**C**)).

**Figure 2 cancers-14-05494-f002:**
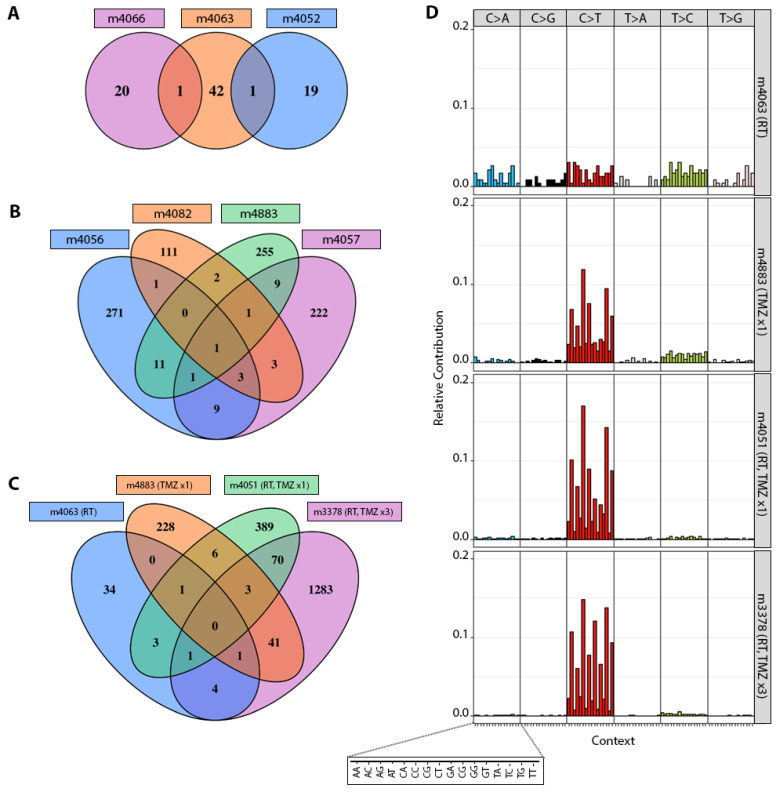
Mutations in GBM6 derivative PDX. (**A**) Venn diagram illustrating numbers of mutated genes in PDX derivatives treated with radiation therapy (m4066, m4063, m4052). (**B**) Numbers of mutated genes in PDX derivatives treated with a single cycle of TMZ (m4056, m4082, m4883, m4057). (**C**) Comparison of numbers of mutated genes in PDX derivatives treated with indicated combinations of RT and TMZ. (**D**) Illustration and comparison of the relative frequency of mutations affecting C and T sites in select GBM6 derivatives treated with the indicated combinations of RT and TMZ.

**Figure 3 cancers-14-05494-f003:**
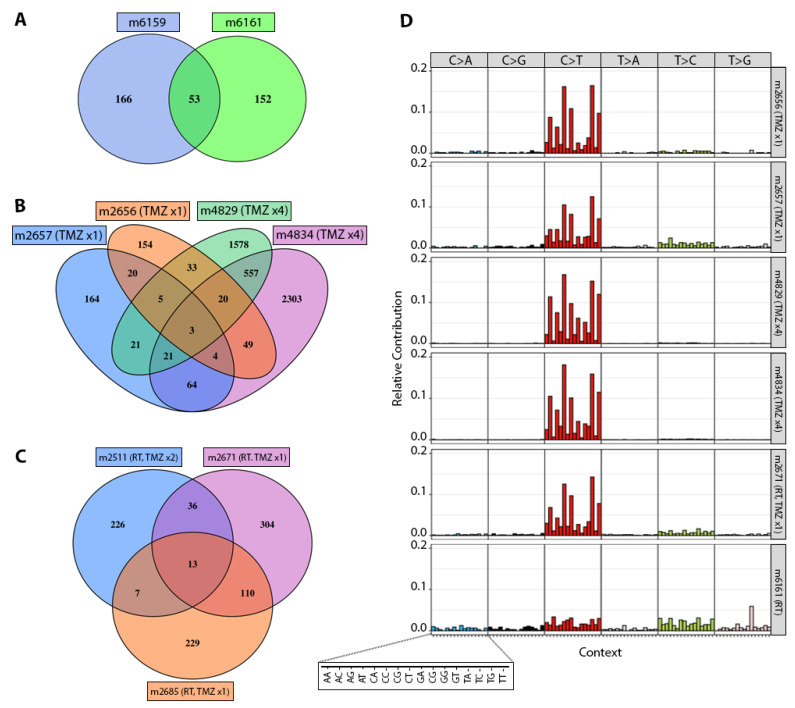
Mutations in GBM12 derivative PDX. (**A**) Venn diagram illustrating numbers of mutated genes in PDX derivatives treated with RT (m6159, m6161). (**B**) Numbers of mutated genes in PDX derivatives treated with a single cycle of TMZ (m2657, m2656) versus four cycles of TMZ (m4829, m4834). (**C**) Numbers of mutated genes in PDX derivatives treated with RT plus a single cycle of TMZ (m2671, m2685) versus RT and 2 cycles of TMZ (m2511). (**D**) Relative frequency of mutations affecting C and T sites in select GBM12 derivatives treated with indicated combinations of RT and TMZ.

**Figure 4 cancers-14-05494-f004:**
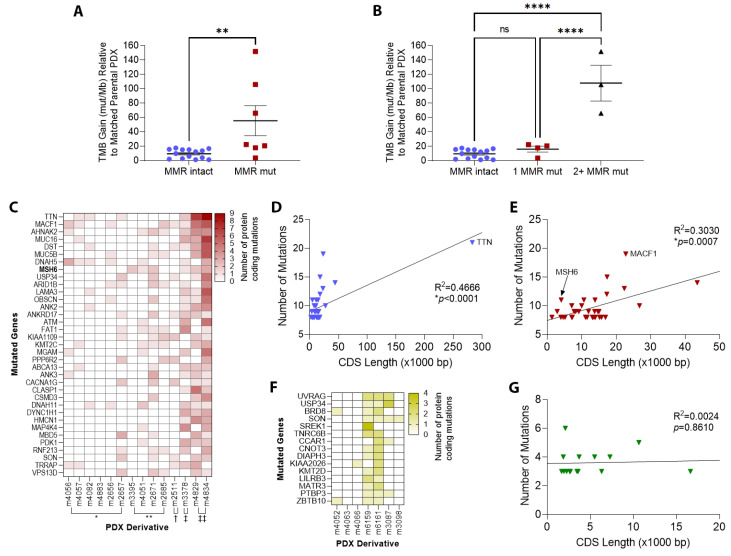
Tumor mutation burden (TMB) and individual gene mutations in derivative PDX. (**A**) TMB gain (relative to matched parental PDX) in derivative PDX with intact DNA mismatch repair (MMR) genes compared to derivative PDX with DNA MMR gene mutations (mut/Mb = mutations per megabase, ** *p* < 0.05). (**B**) TMB gain in derivative PDX with intact DNA MMR genes compared to derivative PDX with a single MMR gene mutation and two or more MMR gene mutations (**** *p* < 0.0001). (**C**) Heat map of the 35 most frequently mutated genes in PDX derivatives treated with TMZ (* TMZ ×1, ** RT+TMZ ×1, ^†^ RT+TMZ ×2, ^‡^ RT+TMZ ×3, ^‡‡^ TMZ ×4). (**D**) Scatterplot illustrating the frequency of protein-coding mutations in genes as a function of gene coding sequence (CDS) length in base pairs (bp). *TTN* has the longest CDS among human genes and was the most frequently mutated in post-TMZ PDX (* *p* < 0.05). (**E**) Frequency of protein-coding mutations as a function of CDS length, with *TTN* eliminated (* *p* < 0.05). (**F**) Heat map of the 15 most frequently mutated genes in PDX derivatives treated with only radiation therapy (RT). (**G**) Frequency of protein-coding mutations as a function of CDS length in PDX derivatives treated with only RT.

**Figure 5 cancers-14-05494-f005:**
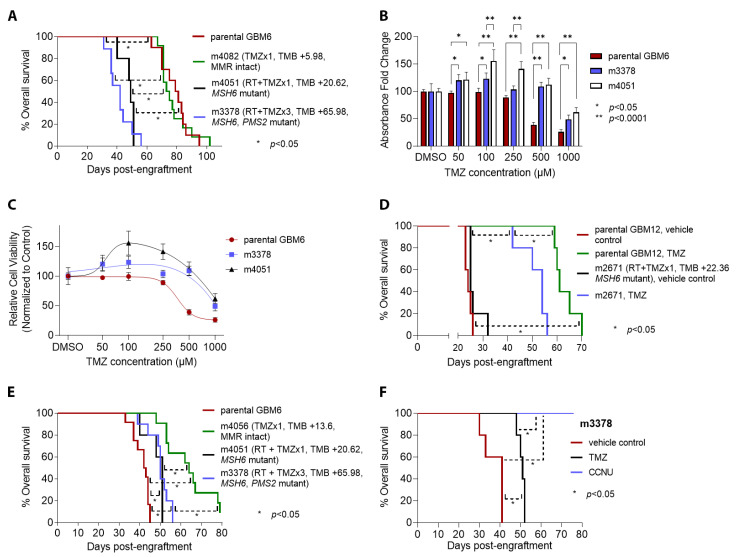
Sensitivity to chemotherapy and radiation in derivative PDX. (**A**) Survival analysis after a single dose of TMZ for mice (*n* = 10 per group) engrafted with parental GBM6 versus derivatives m4082 (post-TMZ ×1, MMR genes intact, low TMB gain), m4051 (post-RT and TMZ ×1, acquired *MSH6* mutation, higher TMB gain), and m3378 (post-RT and TMZ ×3, acquired *MSH6* and *PMS2* mutations, high TMB gain). (**B**) In vitro TMZ sensitivity assays with MTT for parental GBM6 versus derivatives m3378 and m4051 across multiple concentrations of TMZ, with DMSO as vehicle control. (**C**) Best fit dose–response curves to TMZ for parental GBM6 versus m3378 and m4051. (**D**) Survival analysis after a single dose of TMZ (compared to vehicle control) for mice (*n* = 5 per group) engrafted with parental GBM12 versus derivative m2671 (post-RT and TMZ ×1, with acquired *MSH6* mutation). (**E**) Survival analysis after a cycle of RT for mice engrafted with parental GBM6 (*n* = 12), MMR-intact derivative m4056 (*n* = 10), and MMR-deficient derivatives m4051 (*n* = 5) and m3378 (*n* = 10). (**F**) Survival analysis for mice (*n* = 5 per group) engrafted with GBM6 derivative m3378 and treated with a single cycle of TMZ versus a single cycle of CCNU (lomustine), compared to vehicle control (For (**A**,**B**,**D**–**F**), * *p* < 0.05, ** *p* < 0.0001).

**Table 1 cancers-14-05494-t001:** Summary of Parental PDX and Derivative PDX Therapy and Mutations.

Parental PDX	ID *	Therapy **	Genes Mutated	TMB Gain ***	MSH6 ^†^	MSH2 ^†^	MLH1 ^†^	PMS2 ^†^
GBM6	m4052	RT	20	2.10	0	0	0	0
	m4063		44	2.78	0	0	0	0
	m4066		21	1.24	0	0	0	0
GBM12	m6159		219	15.12	0	0	0	0
	m6161		205	15.63	0	0	0	0
GBM43	m3087		48	4.02	0	0	0	0
	m3098		27	1.30	0	0	0	0
GBM6	m4056	TMZ ×1	297	13.60	0	0	0	0
	m4057		249	9.80	0	0	0	0
	m4082		122	5.98	0	0	0	0
	m4883		280	17.59	0	0	0	0
GBM12	m2656		288	12.38	0	0	0	0
	m2657		302	16.67	0	0	0	0
GBM6	m3395	RT+TMZ ×1	78	3.69	2	0	0	0
	m4051		473	20.62	1	0	0	0
GBM12	m2671		463	22.36	2	0	0	0
	m2685		359	15.28	0	0	0	0
	m2511	RT+TMZ ×2	282	17.86	0	0	1	0
GBM6	m3378	RT+TMZ ×3	1403	65.98	1	0	0	1
GBM12	m4829	TMZ ×4	2238	105.78	2	1	0	0
	m4834		3021	151.85	3	1	0	1

* Identification number of individual derivative PDX. ** Therapy regimen for initial intracranial engraftment of PDX (see Figure 1A). *** Increase in tumor mutation burden (TMB), relative to matched parental PDX, measured in mutations per megabase (Mb) of DNA. ^†^ Number of mutations in each DNA mismatch repair (MMR) gene in each derivative PDX. RT = radiation therapy, TMZ = temozolomide cycle.

## Data Availability

All sequencing data included in this manuscript have been submitted to Sequence Read Archive (SRA): https://www.ncbi.nlm.nih.gov/sra/, accession no. PRJNA873104 (accessed on 25 August 2022).

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
