# Peer review of "Modeling Therapy-Driven Evolution of Glioblastoma with Patient-Derived Xenografts"

_cancers, 2022, doi:10.3390/cancers14225494_

Round 1
Reviewer 1 Report
Great work. Not many changes needed.

Reviewer 2 Report
GBM under standard treatment including radiation and temozolomide often recur, which will lead to fatal consequences. Even though Pre-clinical therapies often performed successfully on treatment-naïve mouse models, they usually failed to achieve survival benefit in patients with recurrent GBMs. To solve this issue, McCord et al developed new PDX models using repeated radiation/temozolomide treatment to mimic clinical genotype and phenotype of cancer cell evolution. The manuscript overall shows us meaningful results about therapy-resistant PDX model development based on rounds of therapy exposure. However, with such complicated and time consuming subcutaneous inoculation and treatments, it make this system pretty artificial to me, which is the opposite direction of PDX model generation. I wonder why not derive PDX or CDX based on patients already treated or recurred. The following are some specific points for manuscript.
1. The authors use athymic nude mice as recipient. Nude mice lack T cells however they possess other immune cells, which may lead to immune rejection to xenograft. Could the authors please show the rational of mouse model selection? Will immunodeficient mouse model like NSG be a better choice?
2. Why not generate the PDX model directly using samples from radiation/ TMZ treated patients who recur GBM?
3. In line 96-97, the authors mentioned that tumor dissection method were previously mentioned. Please cite here.
4. In table 1, the authors labeled “intact” or “mutated” on DNA MMR genes. I suggest show specific mutation efficiency to compare with the average gene mutation efficiency in each sample.
Round 2
Reviewer 2 Report
The authors answered my questions. I think the research overall is meaningful considering the key genetic and phenotypic features the model reflected compared to the recurrent GBM in patients. I look forward to seeing the power of this system in drug screening, potential therapy testing et al.